# Attitude towards Intranasal Vaccines and Psychological Determinants: Effects on the General Population in Northern Italy

**DOI:** 10.3390/vaccines11010138

**Published:** 2023-01-07

**Authors:** Paola Boragno, Elena Fiabane, Daiana Colledani, Daniela Dalla Gasperina, Ilaria Setti, Valentina Sommovigo, Paola Gabanelli

**Affiliations:** 1Istituti Clinici Scientifici Maugeri IRCCS, Psychology Unit of Pavia Institute, 27100 Pavia, Italy; 2Istituti Clinici Scientifici Maugeri IRCCS, Department of Physical and Rehabilitation Medicine of Genova Nervi Institute, 16121 Genova, Italy; 3Department of Philosophy, Sociology, Education and Applied Psychology, University of Padua, 35139 Padua, Italy; 4Azienda Socio-Sanitaria Territoriale dei Sette Laghi, Department of Medicine and Surgery, University of Insubria, 21100 Varese, Italy; 5Unit of Applied Psychology, Department of Brain and Behavioural Sciences, University of Pavia, 27100 Pavia, Italy; 6Department of Psychology, Faculty of Medicine and Psychology, Sapienza University of Rome, 00185 Roma, Italy

**Keywords:** intranasal vaccines, vaccine hesitancy, intolerance of uncertainty, persecutory ideation, COVID-19

## Abstract

Little is known about the use of intranasal vaccines, but some studies have shown that this innovative way of administration is preferred over needle injection as it is considered both less painful and intrusive to the body, equally effective, and associated with fewer side effects. This study aimed to investigate specific psychological determinants (intolerance of uncertainty, persecutory ideation, perceived control, vaccine hesitancy) of attitude toward nasal vaccine delivery. A convenience sample including 700 Italian participants took part in this cross-sectional study and completed an online questionnaire. A structural equation model with a latent variable was performed to study the relationship between psychological variables, vaccine hesitancy, and attitude toward nasal vaccine delivery. The results indicate that both a hesitant attitude toward vaccination (*β* = 0.20, *p* = 0.000) and low perceived control (*β* = −0.20, *p* = 0.005) may directly increase preference for nasal administration; furthermore, high levels of persecutory ideation may indirectly influence the propensity for intranasal vaccine. These findings suggest that pharmaceutical companies could implement nasal vaccines and provide detailed information on these vaccines through informational campaigns. Hesitant individuals with low levels of perceived control could more easily comply with these types of vaccines.

## 1. Introduction

Vaccination is considered one of the greatest achievements in public health as it has contributed to reducing the mortality and morbidity that is related to multiple infectious diseases [1]. Immunization programs aim at high vaccination uptake to reduce the prevalence and incidence of vaccine-preventable diseases [2,3]. Recently, countries worldwide have been affected by the coronavirus disease 2019 (COVID-19) pandemic that was caused by SARS-CoV-2, which has been the main source of public health concern [4]. The spread of this virus was mitigated by achieving herd immunity through the vaccine administration.

Nevertheless, vaccine reluctance has been a significant issue since before the COVID-19 era, as a large portion of the population fears possible side effects [5].

The problem of vaccine hesitancy is considered “one of the top ten threats to global health” [6].

To date, COVID-19 vaccines are delivered by intramuscular injection (IM), an invasive method, while many researchers are focusing on producing an effective vaccine to be administered by nasal or oral routes [7].

An intranasal vaccine against live attenuated, cold-adapted influenza virus (LAIV) has been approved in the United States since 2003. Intranasal vaccination against respiratory and gastrointestinal pathogens offers many practical and theoretical advantages [8]. The vaccine is delivered directly to the mucosal surface, and this should enhance mucosal cellular and humoral immune responses [9,10]. Individuals with direct experience with intranasal vaccination tend to favor it over needle injection, as it is considered less painful and less intrusive to the body, equally effective, and associated with fewer side effects [11,12,13]. For example, previous studies have found that the majority of children aged 8–12 years selected mode of administration as one of the most important vaccine attributes, preferring the nasal spray rather than the needle [12]. Even parents’ global opinion and tolerance were judged to be better for intranasal than for intramuscular injection, and they have declared to prefer the LAIV for future vaccination of their children [13]. Additional studies have shown that this mode of administration is also effective and well-accepted among the adult population aged 18–70 years old [14,15].

We questioned whether this innovative vaccine delivery might influence the attitudes of the more hesitant part of the population where hesitant individuals may be more likely to accept nasal spray rather than needle injection as it may be perceived as less intrusive [13]. Several studies showed that vaccine hesitancy is negatively associated with vaccine decision-making [1]. However, no research, to our knowledge, has examined whether hesitancy might lead individuals to favor intranasal versus IM modality of administration. 

A detailed review of the literature on the psychological implications of intranasal vaccines shows a general preference for the intranasal route [13]. The intranasal modality can also potentially increase the effectiveness of vaccination programs as it addresses injection phobia and anxieties, which have been identified as common causes of vaccine hesitancy [16]. 

Vaccine hesitancy is the behavioral delay in acceptance or refusal of vaccines despite the availability of vaccine services [17]. Researchers showed a continuum from acceptance to refusal of all vaccines [18,19,20,21]. Furthermore, psychological factors, such as pain and fear of needles, concerns about side effects, and conspiracy beliefs, may have a significant impact on hesitancy [22,23,24].

Although belief in conspiracy theories is a key factor in paranoid ideation, no direct association has been found between paranoid beliefs and vaccine hesitancy [25]. 

Paranoid individuals have difficulty differentiating objective reality from their sensations and perceptions and tend not to recognize their thoughts as paranoid [26].

For this reason, this research aims to investigate whether paranoid beliefs may indirectly influence hesitancy and attitude toward a nasal vaccine. 

Asides from paranoid beliefs, intolerance of uncertainty (IU) may also affect an individual’s attitude toward vaccination. IU is an “individual’s dispositional incapacity to endure the aversive response triggered by the perceived absence of salient or sufficient information and sustained by the associated perception of uncertainty” [27]. IU is associated with negative psychological effects, including exacerbated perceptions and feelings of risk, avoidance of decision-making, and distrust in scientists and public health guidelines, and these effects vary among individuals, depending on trait-level differences in IU [28,29]. Individuals with a low tolerance for uncertainty and ambiguity may perceive limited benefits of vaccines; hence it is plausible that IU increases vaccine hesitancy [30], particularly toward IM that is perceived as more invasive. 

Individuals with high levels of IU may perceive a weak control exerted over ambiguous and threatening situations and consequently may avoid them to enhance subjectively perceived control over uncertain-threatening environments [31]. In addition, Perrone et al. (2022) demonstrated that the main psychological factor affecting vaccine hesitancy was the perceived lack of control, including the inability to tolerate information that was perceived as ambiguous and uncertain (e.g., potential vaccine side effects) [32]. It is reasonable that individuals with high IU may perceive low control over uncertain external events (i.e., vaccination) and refuse vaccines that are delivered by IM.

This study aimed to investigate specific psychological determinants (intolerance of uncertainty, persecutory ideation, perceived control, vaccine hesitancy) of attitude toward nasal vaccine delivery. Specifically, we explored whether attitude toward nasal vaccine would be (a) positively related to vaccine hesitancy, intolerance of uncertainty, and persecutory ideation and (b) negatively related to perceived control (Figure 1).

## 2. Materials and Methods

### 2.1. Participants and Procedure 

A convenience sample including 700 participants took part in this cross-sectional study. They were recruited from different Italian regions and participated in the research voluntarily and anonymously. They were asked to complete an online questionnaire in Italian language that was administrated using Google Sheets. The participants were contacted through mailing lists and social networks. Moreover, following a snowball sampling procedure, each participant was asked to invite other persons to fill out the online survey. The questionnaire was available only after agreeing to an electronic informed consent indicating the aim of the study, the task duration, and the possibility of withholding the consent to participate in the research at any time. All ethical standards were respected, and the study protocol was approved by the Ethical Committee for Psychological Research of the University of Pavia. 

No specific exclusion criteria were set other than being a native Italian speaker and at least 18 years old. The total sample included 700 Italian individuals (mean age = 41.23; SD = 15.66). Most of them were women (74.1%), employees (64.1%), and living in Northern regions of Italy (80.3%). The majority of them did not work in the healthcare sector (77.7%) and did not suffer from organic diseases (75.4%). A more detailed description of the sample can be found in Table 1.

### 2.2. Measurements 

Vaccine hesitancy and attitude toward intranasal vaccination were investigated by administering 22 items. A questionnaire was specifically developed to investigate individual perceptions of vaccination and intranasal inoculation. To explore vaccine resistance, eleven items were created. Among these, two were yes/no items (e.g., “I have had at least one of the mandatory childhood vaccinations”); six Likert scale items were scored from 1 to 4 (1 = not at all, 2 = a little, 3 = quite a lot, 4 = very much; e.g., “In general, I think vaccines are useful”). Of these, three were reversed and rotated so that high scores were indicators of hesitancy and low scores of vaccine acceptance. There were two items that were multiple-choice (e.g., “Why do you think vaccines may pose a risk? You can select more than one option”) and one was an open-ended question (“What image/word/phrase comes to mind when you think of vaccines?”).

To study attitudes toward an intranasal vaccine, nine items that explore the level of acceptance and preference for LAIV were created. A total of seven Likert scale items scored from 1 to 4 (1 = not at all, 2 = a little, 3 = quite a lot, 4 = very much; e.g., “I worry less about nasal vaccine, because I consider it less invasive for the immune system and for my health.”), where high scores indicate that individuals have positive opinions and tend to prefer intranasal vaccines. Conversely, low scores indicate that subjects do not manifest a preference between nasal and intramuscular vaccines. One question was multiple-choice (“If you had the choice, which would you prefer between the two modes of delivery”), and one Likert scale item scored from 1 to 10 (1 = not at all–10 = very much; “In any vaccination, how much would you be willing to vaccinate yourself with intramuscular/intranasal vaccine on a scale of 1 to 10?”). 

Finally, two questions were asked about the COVID-19 vaccine (e.g., “If the COVID-19 vaccine were administered nasally, I would be more comfortable vaccinating”). This questionnaire is described in the Appendix A (Appendix A).

Intolerance of Uncertainty. The intolerance of uncertainty was measured by administering the Intolerance of Uncertainty Scale-Revised (IUS-R; [33]). The questionnaire is comprised of 12 items (e.g., “Things I don’t know bother me”) scored on a 5-point Likert scale (from 1 = “not at all agree” to 5 = “completely agree”) and assesses two main facets of the construct: Prospective IU and Inhibitory IU. The first facet describes the propensity of individuals to seek information that is aimed to reduce uncertainty, whereas the second evaluates avoidance-oriented responses to uncertainty. High scores to the scale indicate the subjective inclination to perceive situations of uncertainty as threatening and the willingness to implement strategies to reduce the discomfort that is caused by them (e.g., seeking information, and avoiding situations of uncertainty). Studies in the Italian context supported the satisfactory psychometric properties of the scale and the use of a total scale score [31]. In the present study, the reliability of this scale was α = 0.88.

Perceived control. The Perceived Control over Events (PCE) subscale of the Italian version of the Anxious Control Questionnaire scale (ACQ; [34]) was used to measure perceived control. The PCE scale includes 16 items (e.g., “I’m usually able to easily avoid threats”) scored on a 5-point Likert scale (from 1 = “strongly disagree” and 5 = “strongly agree”). High scale scores describe individuals who perceive they have adequate control over external events, while low scores describe individuals who feel poor control over events and may indicate anxiety disorders [34]. The literature suggests the adequate validity and reliability of the scale (ACQ; [34]). In the present work, item 30 (i.e., “I want to avoid the conflicts due to my inability to resolve them”) showed negative correlations with all the remaining items, consequently, it was eliminated from the computation of the total score. Excluding item 30, the reliability of the scale was α = 0.77.

Persecutory ideation. The Persecutory Ideation Questionnaire (PIQ; [35]) was administered to evaluate the individuals’ tendency to be suspicious and to perceive even neutral situations as threatening. Unfortunately, this questionnaire has not yet been translated and validated in Italian languages, thus PIQ items were translated by three native Italian speakers. Then, the items were back-translated into English by a native speaker. Finally, Italian and English native speakers compared the back-translated version with the original version of the scale. The questionnaire includes ten items (e.g., “Some people harass me persistently”) scored on a 5-point Likert scale (from 0 = “completely false” to 4 = “completely true”). The higher the scale scores the higher the individuals’ levels of persecutory ideations. The literature denotes adequate psychometric properties for this instrument in both the clinical (α = 0.90) and non-clinical samples (α = 0.87, [35]). In the present study internal consistency, represented by Cronbach’s α and composite reliability, was excellent (α = 0.92; CR = 0.91). Convergent validity, calculated by average variance extracted, was good (AVE = 0.53). In sum, the PIQ has been shown to have good psychometric properties in this sample, and, therefore, persecutory ideation can be measured reliably with the Italian version of the PIQ in general population.

### 2.3. Statistical Analyses

The hypothesized relationships between constructs were tested using a structural equation model with latent variables (see Figure 1). In the model, IU and persecutory ideations were the independent variables, perceived control and vaccine hesitancy were two mediating variables operating in series, and the preference for nasal vaccination was the dependent variable. All the constructs were measured using two (preference for intranasal inoculation, vaccine hesitancy, and persecutory ideations) to three (IU and perceived control) parcels, built using the random assignment method [36]. This method, improving the ratio between the sample size and the number of estimated parameters, allows for obtaining more stable estimates [36,37,38]. A total of five exogenous covariates were also included in the model to control for the effects of gender, age, education level, work in the healthcare sector, and organic diseases. The structural model was evaluated after the measurement model was accepted and the potential overlap between constructs was verified. All the direct paths were estimated, and the significance of indirect effects was evaluated employing bootstrapping procedures (5000 resamples) and the 95% bias-corrected confidence interval. The model was estimated using the maximum likelihood (ML) estimator and its fit was evaluated using several indices: χ^2^, root mean square error of approximation (RMSEA), comparative fit index (CFI), and standardized root mean square residual (SRMR). The adequacy of a model is supported by non-significant χ^2^ values, RMSEA lower than 0.06 (0.06 to 0.08, for a reasonable fit), CFI close to 0.95 (0.90 to 0.95, for a reasonable fit), and SRMR less than 0.08 [36]. Soper’s (2020) calculator for structural equation models indicated that with 17 observed (the 12 parcels plus the 5 covariates) and 5 latent variables, the minimal sample size that is required to reach a power of 0.80, with a probability level of 0.05 and an effect size of 0.20 (between small, f = 0.10, and medium, f = 0.30), is of 376 respondents

## 3. Results

### 3.1. Structural Equation Model with Latent Variables

First, the goodness-of-fit of parcel measurement methodology was tested. The fit indices were excellent [χ^2^(44) = 80.180, *p* < 0.01; RMSEA = 0.03, RMSEA 95% CI [0.02, 0.05]; CFI = 0.99; TLI = 0.99; SRMR = 0.02] and the range of factor loadings between parcels and latent variables was between 0.68 and 0.96 (see Figure 2), indicating good strength of association between aggregate items and latent factors. The composition of parcels can be consulted in the section “Appendix A” (see Appendix A). Subsequently, the structural equation model with latent variables was constructed: latent variables persecutory ideation and intolerance of uncertainty were handled as independent variables, perceived control over events as first-level moderators, the factor vaccine hesitancy was treated as a second-level serial moderator, and the construct attitude toward nasal vaccine as a dependent variable. The final model is represented in Figure 2. 

Initially, two models were compared: a model with the covariates gender, age, education level, being (vs. being not) employed as a healthcare provider, and having (vs. having not) had previous organic diseases and a model without covariates. This procedure was performed to estimate the effects between latent variables while controlling for covariates. It was observed that the strength of associations between variables did not change in the models that were examined, and, therefore, it was determined to keep the most parsimonious model without socio-demographic control variables. As the majority of effects of socio-demographic variables on the latent constructs were very small in size and not statistically significant, multigroup models were not constructed to compare differences in latent constructs between socio-demographic variables (the results of the model with socio-demographic covariates can be consulted in the section “Appendix A”, Appendix A).

Fit indices of the model without covariates were very good [χ^2^(44) = 80.180, *p* < 0.01; RMSEA = 0.03 95% RMSEA CI [0.02, 0.05]; CFI = 0.99; TLI = 0.99; SRMR = 0.02]. Consequently, the interpretation of direct and indirect effects was carried out. Degrees of Freedom were 46, and the percentage of dependent latent variables’ variance explained (R^2^) can be seen in Table 2.

### 3.2. Direct Effects

The perceived control variable manifested a significant and moderate-sized negative direct effect (*β* = −0.20, *p* = 0.005) on the variable attitude toward nasal vaccine, showing that perception of having adequate control over external events is positively related to a reduction in preference for vaccines intranasal administration.

The variable hesitancy showed a significant and moderate-sized positive direct effect (*β* = 0.20, *p* = 0.000) on the attitude toward nasal vaccine variable, suggesting that a hesitant attitude toward vaccination increases preference for nasal administration. Intolerance of uncertainty (*β* = 0.11, *p* = 0.095) and persecutory ideation (*β* = −0.01, *p* = 0.900) exhibited no significant effect on the construct attitude towards nasal vaccination.

Perceived control manifested a significant and large negative direct effect (*β* = −0.33, *p* = 0.000) on vaccine hesitancy, indicating that the perception of having adequate control over external events can generate favorable attitudes toward vaccination. 

Intolerance of uncertainty showed a negative direct effect of moderate size (*β* = −0.24, *p* = 0.000) on vaccine hesitancy, suggesting that a high level of intolerance of uncertainty makes people less hesitant. 

The variable persecutory ideation presented no significant direct effect on the vaccine hesitancy construct (*β* = 0.11, *p* = 0.124). 

The latent variable persecutory ideation manifested a significant and large negative direct effect (*β* = −0.23, *p* = 0.000) on the perceived control variable, revealing that a high level of paranoid ideation may reduce the perception of having adequate control over external events. 

The latent variable intolerance of uncertainty demonstrated a significant and large negative direct effect (*β* = −0.57, *p* = 0.000) on the perceived control variable, indicating a high level of intolerance of uncertainty may reduce the perception of having adequate control over external events. To observe the graphical representation of direct effects, see Figure 2. For non-standardized direct effects, see Table 3.

### 3.3. Indirect Effects

Only significant indirect effects will be commented on below. 

The perceived control over events variable manifested a significant positive indirect mediating effect between persecutory ideation and vaccine hesitancy (*β* = 0.08, *p* = 0.001). 

The perceived control over events variable manifested a positive and significant indirect mediating effect between intolerance of uncertainty and vaccine hesitancy (*β* = 0.19, *p* = 0.000). 

Vaccine hesitancy exerted a positive and significant indirect mediating effect between control and attitude toward nasal vaccine (*β* = 0.07, *p* = 0.004).

Perceived control over events exerted a positive and significant indirect mediating effect between the persecutory ideation and attitude toward nasal vaccine (*β* = 0.05, *p* = 0.014).

Perceived control and vaccine hesitancy were positive and significant serial mediators of the association between persecutory ideation and attitude towards the nasal vaccine (*β* = 0.02, *p* = 0.012).

Perceived control over events exerted a positive and significant indirect mediating effect between intolerance of uncertainty and attitude towards a nasal vaccine (*β* = 0.11, *p* = 0.006).

Vaccine hesitancy proved to be a negative and significant mediator of the direct association between the construct intolerance of uncertainty and attitude towards the nasal vaccine (*β* = −0.05, *p* = 0.005).

Perceived control and vaccine hesitancy were positive and significant serial mediators of the relationship between intolerance of uncertainty and attitude towards the nasal vaccine (*β* = 0.04, *p* = 0.004).

For non-standardized effects, see Table 4.

## 4. Discussion

This study aimed to investigate specific psychological determinants of attitude toward the nasal vaccine. Few scientific studies have investigated the association between psychological variables, vaccine hesitancy, and attitudes towards a nasal vaccine [7,8,9,13,39]. This study aims to fill this gap by studying the relationship between psychological variables (intolerance of uncertainty, perceived control, and paranoid ideation), vaccine hesitancy, and attitude toward an intranasal vaccine.

We found that an important additional direct effect is the significant relationship between vaccine hesitancy and attitude towards a nasal vaccine. Specifically, high levels of hesitancy make individuals more favorable to nasal spray. The spray may be perceived as a more circumscribed, limited, and controllable external agent; in contrast, the needle could be perceived as an aggressive external agent that radiates throughout the body and is more invasive to the immune system. The non-invasive mucosal vaccine delivery has been previously studied because of its clinical (e.g., eliminates needle-associated risks) and practical (e.g., can be self-administered) advantages [9,39] but, to the best of our knowledge, no previous studies have investigated its relationship with psychological dimensions such as hesitancy. Therefore, the results of this study should be considered as preliminary and future studies exploring this issue are encouraged. 

IU manifested a significant negative direct effect on hesitancy where individuals with high traits of IU tended to be inclined to vaccinate; this result is inconsistent with the study of Gillman et al. where a lower tolerance of ambiguity was associated with lower intentions to get vaccinated [30]. These divergent results may be explained by the timing of the data collection. Gillman et al.’s study was conducted in the earlier months of the pandemic in 2020 when no effective treatments or vaccines against COVID-19 were available, while our study was conducted in an advanced phase of vaccination campaign. Our findings might indicate that illness is experienced as an uncertain and uncontrollable risk, more than the vaccine itself. The non-significant direct effect between IU and attitude toward intranasal vaccine reinforces this interpretation where the mode of vaccine administration is indifferent because the goal is to protect oneself and to contain the sense of uncertainty that is associated with the risk of getting infected. Vaccines have become the container of anxiety, represented by the risk of getting sick.

IU exerted a contrasting effect on vaccine hesitancy. On one hand, intolerance of uncertainty positively influences subjects’ attitude towards vaccination; on the other hand, if perceived control over events is involved, this effect reverses, meaning that intolerance of uncertainty makes people more hesitant. Therefore, individuals who have only low internal self-efficacy tend to vaccinate, as the vaccine contains this internal suspension and tension. In contrast, low internal and external self-efficacy leads to a stalemate, in which the vaccine decision is postponed and delayed. 

A direct relationship was not found among the persecutory ideation and vaccine hesitancy and attitude toward intranasal vaccine, confirming the results of Andrade’s study [25] who found a low correlation between paranoid ideation and vaccine hesitancy in a sample of Venezuelan university students. However, as assumed, persecutory ideation has shown to exert a positive indirect effect on both the vaccine hesitancy variable and the intranasal vaccine preference variable due to the mediating role of perceived control. Individuals with persecutory traits usually perceive the outside world as uncontrollable and threatening, feeling under attack and, consequently, refusing vaccination. In addition, people with paranoid traits may perceive the nasal spray as more controllable and less threatening, able to reduce worries and tension compared to the needle.

Perceived control over events showed a direct negative effect on vaccine hesitancy where there was a greater the feeling of exerting adequate outward control, the lower the hesitant attitude. Control corresponds to the ability to master and manage information from the outside world. So, the more an individual feels to possess this ability, the more confident and calm he or she will be in dealing with vaccination. These results are in line with the qualitative study by Perrone et al. who suggested that the main theme of hesitancy was the perceived lack of control [32]. Perceived control over events in this study also exerted a significant direct effect on attitude toward intranasal vaccine. Specifically, subjects who perceived to manifest good outward control did not feel threatened by the needle, did not experience the injection as distress, and did not favor intranasal vaccine. Given that high levels of perceived control seem to be a key determinant of accepting attitudes toward nasal vaccines, focusing on boosting perceived control may be helpful for promoting vaccination acceptance. To this aim, mindfulness interventions may enhance individuals’ flexibility in responding to changing and uncertain environments and then increase their perceived control over events. Additionally, since intolerance of uncertainty and persecutory ideation can compromise perceived control, it could be helpful to formulate communication strategies for supporting tolerance of threats and uncertainties and persecutory thoughts, especially during pandemic times [25].

### 4.1. Limitations

The results of this study should be interpreted in light of some limitations. Firstly, because of the cross-sectional nature of our study, causal relationships cannot be inferred; therefore, a longitudinal design would be necessary to explore the causality of these relationships. Future longitudinal and experimental studies will be necessary to provide a better methodological framework in which to test our hypothesis. Secondly, this research merely relies on self-reported measures and thus suffers the limitations from such a methodology (e.g., recall bias, social desirably). Future research should collect multisource and multimethod data. Thirdly, selection bias cannot be ruled out due to respondents’ voluntary participation in this study. Future work could include an incentive for participants to encourage data collection from a more representative sample to minimize bias and improve the generalizability of the results. 

Fourthly, the authors tested an innovative model which introduces a poorly explored variable (i.e., attitudes towards intranasal vaccines) using an ad hoc questionnaire that was specifically created for this study. This research is explorative and future studies are needed to support this model and this instrument. Finally, it was outside the purposes of this study to test whether the conducted structural equation models would differ based on socio-demographic variables. Future studies should replicate our findings by collecting data on larger and better-balanced samples to test differences across groups using multi-group analyses and to identify which potential unmeasured moderating variables might intervene in the analyzed relationships.

### 4.2. Conclusions

To date, evidence about psychological factors and personal motivations that are implicated in the personal attitudes towards intranasal vaccination is limited. The present study started filling this gap by focusing on some psychological variables—including vaccine hesitancy, intolerance of uncertainty, perceived control, and persecutory ideation—affecting propensity for intranasal or intramuscular vaccine intentions.

Our work explored the complex phenomenon of vaccine hesitancy and focused on how certain psychological dimensions might affect preference for vaccine delivery.

Although the results of this study do not permit us to conclude whether or not hesitant individuals prefer intranasal vaccine, the findings indicate that hesitancy and specific psychological variables, paranoia and perceived control, may affect attitudes toward nasal vaccines. As demonstrated by the pre-existing literature, intranasal immunization possesses different advantages. It has much higher patient compliance; it can be self-administered, eliminating the need for specialized personnel and significantly reducing the cost of mass vaccination; it is easy to use; safe; multifunctional; and can be distributed quickly [39].

In the light of benefits and greater adherence, pharmaceutical companies could implement nasal vaccines and provide detailed information on these vaccines through informational campaigns. In fact, the most hesitant individuals could more easily comply with these types of vaccines. Vaccine hesitancy is indeed a complex phenomenon, and these findings can improve the possibility to tailor efficacious vaccine-related communications that drive people’s acceptance of vaccines [40]. Future research could explore attitudes toward nasal vaccines by implicit measures, such as the Implicit Association Test (IAT), to examine the underlying cognitive associations and beliefs.

## Figures and Tables

**Figure 1 vaccines-11-00138-f001:**
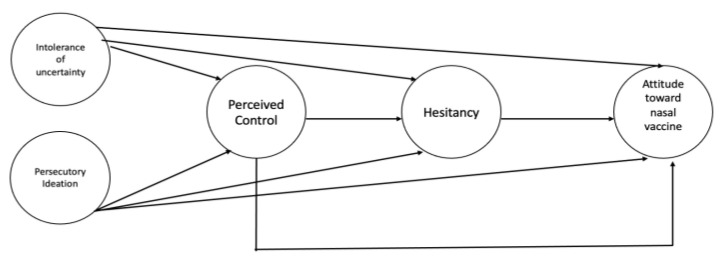
Conceptual model. These variables are latent and measured through the parcels method. The relationships among these variables will be tested while controlling for the following socio-demographic variables: gender, age, educational level, health professional worker, and organic disease.

**Figure 2 vaccines-11-00138-f002:**
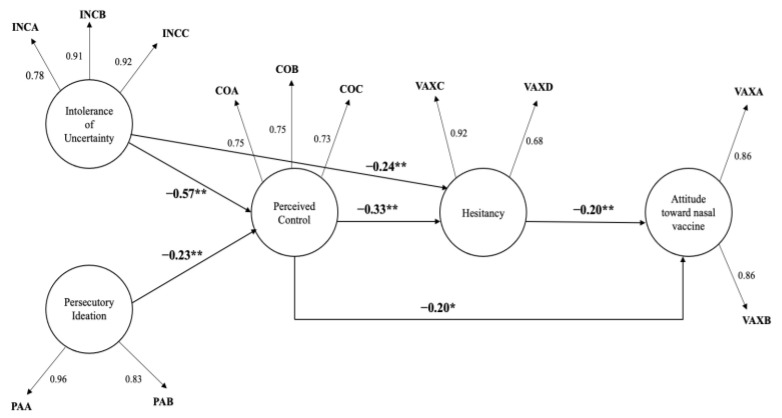
Graphical representation of the structural equation model with latent variables. * *p* < 0.05; ** *p* < 0.01.

**Table 1 vaccines-11-00138-t001:** Descriptive statistics of the sample (n = 700).

	% (n)
Gender	
Women	74.1 (519)
Men	25.7 (180)
Other	0.1 (1)
Education level	
Primary and middle school	4.0 (28)
High school	29.1 (204)
Bachelor or Masters Degree	47.7 (334)
Post-university	19.1 (134)
Occupation	
Unemployed	6.4 (45)
Employed	64.1 (449)
Retired	9 (63)
Student	20.4 (143)
Geographic location	
Northern Regions (Lombardia, Liguria, Piedmont, Aosta Valley, Emilia Romagna, Friuli-Venezia Giulia, Trentino-Alto Adige, Veneto)	80.3 (562)
Central Regions (Lazio, Tuscany, Umbria, Marche)	6.4 (45)
Southern Regions and Islands (Abruzzo, Basilicata, Campania, Molise, Apulia, Sardinia, Sicily)	13.3 (93)
Healthcare worker	
Yes	22.3 (156)
No	77.7 (544)
Organic diseases	
No	75.4 (528)
Yes	24.6 (177)

**Table 2 vaccines-11-00138-t002:** R-square of latent independent variables.

		R-SQUARE		
Latent Variables	Estimate	S.E.	Est./S.E.	*p*-Value
Hesitancy	0.085	0.034	2.534	0.011
Attitude	0.137	0.032	4.339	0.000
Control	0.487	0.038	12.767	0.000

**Table 3 vaccines-11-00138-t003:** Non-standardized direct effects.

Non-Standardized Direct Effects				
	P.E.	95% CI	S.E.	*p*-Value
Perceived control → Attitude toward nasal vaccine	−0.215	[−0.379, −0.067]	0.08	0.006
Vaccine hesitancy → Attitude toward nasal vaccine	0.216	[0.377, 0.095]	0.07	0.002
Perceived control → Vaccine hesitancy	−0.322	[−0.170, − 0.483]	0.08	0.000
Intolerance of uncertainty → Vaccine hesitancy	−0.243	[−0.114, −0.371]	0.07	0.000
Persecutory Ideation → Perceived control	−0.197	[−0.270, − 0.120]	0.04	0.000
Intolerance of uncertainty → Perceived control hesitancy	−0.603	[−0.709, −0.507]	0.05	0.000

Note. Only Significant direct effects have been reposted. P.E. = Point Estimate; CI = 95% Confidence Interval.

**Table 4 vaccines-11-00138-t004:** Non-standardized indirect effects.

Non-Standardized Indirect Effects		
	P.E.	95% CI
Persecutory ideation → **Perceived control** → Vaccine hesitancy	0.06	[0.109, 0.030]
Intolerance of uncertainty → **Perceived control** → Vaccine hesitancy	0.19	[0.296, 0.107]
Perceived control → **Vaccine hesitancy** → Attitude toward nasal vaccine	0.07	[0.138, 0.030]
Persecutory ideation → **Perceived control** → Attitude toward vaccine	0.04	[0.014, 0.083]
Persecutory ideation → **Perceived control** → **Vaccine hesitancy** → Attitude toward nasal vaccine	0.01	[0.005, 0.031]
Intolerance of uncertainty → **Perceived ontrol** → Attitude toward nasal vaccine	0.13	[0.044, 0.235]
Intolerance of uncertainty → **Vaccine hesitancy** → Attitude toward nasal vaccine	−0.05	[−0.107, 0.022]
Intolerance of uncertainty → **Perceived control** → **Vaccine hesitancy** → Attitude toward nasal vaccine	0.13	[0.021, 0.275]

Note: Only significant indirect effects have been reported. Variables that exert indirect effects are shown in bold. P.E. = point estimate; 95% CI = 95% confidence interval. Perceived control = perceived control over external events.

## Data Availability

The data that are presented in this study are available on request from the corresponding author. The data are not publicly available due to privacy.

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
