# Peer review of "Attitude towards Intranasal Vaccines and Psychological Determinants: Effects on the General Population in Northern Italy"

_vaccines, 2023, doi:10.3390/vaccines11010138_

Round 1
Reviewer 1 Report
Study: Authors conducted an online survey to get perceptions about the intranasal vaccines by convenience sampling of 700 participants.
Title of the study: since it only included the Italian population, this should be addressed in the title ; Like " Attitude towards intranasal vaccines and psychological determinants of Italian population: which effects on the general population? " Also the majority of the participant is from one geographic area ( 80%) authors might consider adding it in the title.
Abstract: looks good
Introduction: overall ok, can authors talk more about which population is more receptive of nasal vaccination? Like adults or children, if there is any data on it ? Authors have included only 18 years and above population but kids might be more receptive to Nasal vaccinations. "Just a suggestions"
Material and method: Please mention in what language the survey was administered and please include ur survey tool in the supplementary section
Measurements;
The authors included a 4-point Likert scale. Please explain why did they not include the neutral point.
They have use likert scale from 1-4 to 1-10 , It might be more useful to use the same scale across the board like 5 point likert scale but it can not be changed now so mention it in the limitations.
Results :
Discussion: well done with the proper limitation section. Might want to add it as a subsection and also can add a subsection of the conclusion.
Author Response
Point-by-point response to the reviewer’s comments
POINT 1
English language and style
( ) English very difficult to understand/incomprehensible
( )Extensive editing of English language and style required
( ) Moderate English changes required
(x) English language and style are fine/minor spell check required
( ) I don't feel qualified to judge about the English language and style
AUTHORS’ ANSWER
We have carefully revised English language.
POINT 2
Comments and Suggestions for Authors
Study: Authors conducted an online survey to get perceptions about the intranasal vaccines by convenience sampling of 700 participants.
Title of the study: since it only included the Italian population, this should be addressed in the title ; Like " Attitude towards intranasal vaccines and psychological determinants of Italian population: which effects on the general population? " Also the majority of the participant is from one geographic area (80%) authors might consider adding it in the title.
AUTHORS’ ANSWER
Thank you very much for your suggestion. We agreed with your comment and we revised the title in order to specify the geographic area: “Attitude towards intranasal vaccines and psychological determinants: effects on the general population in Northern Italy”
POINT 3
Abstract: looks good
AUTHORS’ ANSWER
Authors thanks the reviewer for this evaluation.
POINT 4
Introduction: overall ok, can authors talk more about which population is more receptive of nasal vaccination? Like adults or children, if there is any data on it? Authors have included only 18 years and above population but kids might be more receptive to Nasal vaccinations. "Just a suggestions"
AUTHORS’ ANSWER
Authors thank reviewer for this relevant suggestion. We agree that it is an important topic and we have added in the introduction a specific sentence (line 62-68, page 2); we also cited some interesting papers (see for example, reference 12, 13, 20, 21, 23)
POINT 5
Material and method: Please mention in what language the survey was administered and please include ur survey tool in the supplementary section
AUTHORS’ ANSWER
We have added in the text (line 125, page 3) the language of the survey. Furthermore, as you suggested, we have included the questionnaire that we have ad-hoc created to measure “Vaccine hesitancy and attitude toward intranasal vaccination” (see Table S1). The other questionnaires used in this study (Intolerance of Uncertainty Scale-Revised (IUS-R); Persecutory Ideation Questionnaire (PIQ); Anxious Control Questionnaire scale (ACQ)) were validated and available in the references section of the paper.
POINT 6
Measurements;
The authors included a 4-point Likert scale. Please explain why did they not include the neutral point.
AUTHORS’ ANSWER
We made this choice because asymmetric Likert scale offers less choices of neutrality; indeed, asymmetric scale in some cases forces choices where the researcher aims to reduce attitudes of neutrality (see for example, Tsang KK. The use of midpoint on Likert scale: The implications for educational research. Hong Kong Teachers Centre Journal. 2012, 11:121-130; Malhotra NK. Questionnaire Design and Scale Development. In: Grover R, Vriens M, editors. The Handbook of Marketing Research. California: Sage Publications, Inc; 2006).
POINT 7
They have used likert scale from 1-4 to 1-10, It might be more useful to use the same scale across the board like 5 point likert scale but it can not be changed now so mention it in the limitations.
AUTHORS’ ANSWER
Thank you for this comment. We excluded the item with 1-10 likert scale from the statistical analyses; parcels were created using all items on 4-point Likert scale (for details see Table S2 in Supplementary Materials Section).
However, we have added in the limitation section that we used an ad-hoc questionnaire specifically created in this research to measure attitudes towards intranasal vaccine, and that further studies are needed to confirm the validity of this instrument (line 411-414, page 11).
POINT 8
Results:
Discussion: well done with the proper limitation section. Might want to add it as a subsection and also can add a subsection of the conclusion.
AUTHORS’ ANSWER
We followed your suggestion and we have modified the discussion accordingly (page 11-12, line 399-444)
The discussion section of the paper now has a subsection “Limitations” and a subsection “Conclusions”.
Reviewer 2 Report
Thank you for sending me the manuscript for review. The article is done with good scientific approach and the results are interesting. However, some modifications are needed before making any decision.
1. The abstract section can be improved by presenting major findings.
2. To my liking, the introduction can be made more concise with clear aim and objectives.
3. The discussion need some refinement with more inclusion of compare/contrast with previous findings.
4. I think separate conclusion section would be more good.
5. Study limitations must be included.
Author Response
Point-by-point response to the reviewer’s comments
POINT 1
English language and style
( ) English very difficult to understand/incomprehensible
( ) Extensive editing of English language and style required
( ) Moderate English changes required
(x) English language and style are fine/minor spell check required
( ) I don't feel qualified to judge about the English language and style
AUTHORS’ ANSWER
We have carefully revised English language.
POINT 2
Comments and Suggestions for Authors
Thank you for sending me the manuscript for review. The article is done with good scientific approach and the results are interesting. However, some modifications are needed before making any decision.
- The abstract section can be improved by presenting major findings.
AUTHORS’ ANSWER
Thank you for this comment. We have revised the abstract accordingly in order to improve major findings (line 29-31, page 1).
POINT 3
- To my liking, the introduction can be made more concise with clear aim and objectives.
AUTHORS’ ANSWER
We have significantly revised the aims in order to make the introduction more concise and the objectives clearly defined as follows:
“This study aimed to investigate specific psychological determinants (intolerance of uncertainty, persecutory ideation, perceived control; vaccine hesitancy) of attitude toward nasal vaccine delivery. Specifically, we explored whether attitude toward nasal vaccine would be a) positively related to vaccine hesitancy, intolerance of uncertainty and persecutory Ideation and b) negatively related to perceived control (Figure 1)”.
POINT 4
- The discussion need some refinement with more inclusion of compare/contrast with previous findings.
AUTHORS’ ANSWER
We have significantly revised the discussion in order to add comparison with literature. Here some examples:
page 10, Line 344-349:
“The non-invasive mucosal vaccine delivery has been previously studied because of its clinical (e.g., eliminates needle-associated risks) and practical (e.g. can be self-administered) advantages [9, 39] but, to the best of our knowledge, no previous studies investigated its relationship with psychological dimensions such as hesitancy; therefore, results of this study should be considered as preliminary and future studies exploring this issue are encouraged.”
Page 10, Line 351-356:
“this result is inconsistent with the study of Gillman and colleagues where lower tolerance of ambiguity was associated with lower intentions to get vaccinated [30]. These divergent results may be explained by the timing of the data collection: Gillman and colleagues’ study was conducted in the earlier months of the pandemic in 2020 when no effective treatments nor vaccines against COVID-19 were available, while our study was conducted in an advanced phase of vaccination campaign.”
Page 10, Line 370-373:
“Direct relationship was not found among the persecutory ideation and vaccine hesitancy and attitude toward intranasal vaccine, confirming results by Andrade's study [25] who found a low correlation between paranoid ideation and vaccine hesitancy in a sample of Venezuelan university students.”
Page 10, Line 384-386:
“These results are in line with the qualitative study by Perrone and colleagues who suggested that the main theme of hesitancy was the perceived lack of control [32].”
POINT 5
- I think separate conclusion section would be more good.
AUTHORS’ ANSWER
Thank you for this suggestion. We have created a subsection “Conclusions” (page 11-12)
POINT 6
- Study limitations must be included.
AUTHORS’ ANSWER
Thank you for this suggestion. We have created a subsection “Limitations” (page 11)
Reviewer 3 Report
The article discusses a very important issue, especially in the light of the recent COVID-19 pandemic. Vaccine hesitancy is a major obstacle to achieving proper vaccination and herd immunity among the general public. This type of research has been gaining a lot of attention in the recent years. I believe it is crucial to understand the attitude of people towards vaccines and further investigate the determinants leading to hesitancy and acceptance to vaccines. Furthermore, the focus on intranasal vaccines was of significant importance; the lack of research on this matter warrants further investigation.
The article is well structured and well-written, although brief language editing is recommended. The article was not very simple and easy to comprehend, partly due to the complex behavioral and psychological aspects discussed. However, it is very innovative how persecutory ideation, intolerance of uncertainty, and perceived control influence vaccine hesitancy and in turn affect one’s attitude towards intranasal vaccines. The methods are reproducible and understandable, even though the questionnaires used were very long. The use of different revised scores was adept and made understanding and analyzing the data much easier. The factors studied are numerous, rendering the results very confusing.
Comments:
- the title: Attitude toward intranasal vaccines and psychological determinants: which effects on the general population? in the following sentence: “which effects on the general population?”, can you do a language check.
- Table 1: regarding the geographic location, the percentages do not add up to 100%; 80.3% and13.3% add up to 93.6%, suggesting a gap in the results.
- Line 312: check the alignment.
The conclusion is consistent with the arguments discussed in the manuscript. The outcome of this study is beneficial; understanding the psychology behind vaccine hesitancy can help us combat this issue and therefore come up with alternative ways to vaccinate, for instance intranasal vaccines. I believe we still require further work on this matter and increase awareness regarding intranasal vaccines.
I would to thank the authors for this paper, it was a pleasure to read and review.
Author Response
Point-by-point response to the reviewer’s comments
The manuscript has been revised according to the suggestions and comments of the reviewer. Here is the point-by-point response to the comments.
POINT 1
English language and style
( ) English very difficult to understand/incomprehensible
( ) Extensive editing of English language and style required
( ) Moderate English changes required
(x) English language and style are fine/minor spell check required
( ) I don't feel qualified to judge about the English language and style
AUTHORS’ ANSWER
We have carefully revised English language.
Comments and Suggestions for Authors
The article discusses a very important issue, especially in the light of the recent COVID-19 pandemic. Vaccine hesitancy is a major obstacle to achieving proper vaccination and herd immunity among the general public. This type of research has been gaining a lot of attention in the recent years. I believe it is crucial to understand the attitude of people towards vaccines and further investigate the determinants leading to hesitancy and acceptance to vaccines. Furthermore, the focus on intranasal vaccines was of significant importance; the lack of research on this matter warrants further investigation.
The article is well structured and well-written, although brief language editing is recommended. The article was not very simple and easy to comprehend, partly due to the complex behavioral and psychological aspects discussed. However, it is very innovative how persecutory ideation, intolerance of uncertainty, and perceived control influence vaccine hesitancy and in turn affect one’s attitude towards intranasal vaccines. The methods are reproducible and understandable, even though the questionnaire used were very long. The use of different revised scores was adept and made understanding and analyzing the data much easier. The factors studied are numerous, rendering the results very confusing.
Authors thank the reviewer for his evaluation; we paid attention to all your comments and we hope that this revised version of the paper fulfills your requests.
POINT 2
Comments:
- the title: Attitude toward intranasal vaccines and psychological determinants: which effects on the general population? in the following sentence: “which effects on the general population?”, can you do a language check.
AUTHORS’ ANSWER
Thank you very much for your suggestion. We checked the language and we have modified the title according to your comment and also in agreement with reviewer 1. The revised title is the following:
“Attitude towards intranasal vaccines and psychological determinants: effects on the general population in Northern Italy”
POINT 3
- Table 1: regarding the geographic location, the percentages do not add up to 100%; 80.3% and 13.3% add up to 93.6%, suggesting a gap in the results.
AUTHORS’ ANSWER
We thank a lot the reviewer for this comment since we found an error in the geographic location variable: we forgot to insert participants from the “Central Regions” of Italy. Now the results are corrected.
POINT 4
- Line 312: check the alignment.
AUTHORS’ ANSWER
Thank you, the alignment is fixed.
The conclusion is consistent with the arguments discussed in the manuscript. The outcome of this study is beneficial; understanding the psychology behind vaccine hesitancy can help us combat this issue and therefore come up with alternative ways to vaccinate, for instance intranasal vaccines. I believe we still require further work on this matter and increase awareness regarding intranasal vaccines.
I would to thank the authors for this paper, it was a pleasure to read and review.
We sincerely appreciate the reviewer's comments and we are grateful that the relevance of this topic has been enhanced.
Round 2
Reviewer 2 Report
Accept